# Tannic Acid with Antiviral and Antibacterial Activity as A Promising Component of Biomaterials—A Minireview

**DOI:** 10.3390/ma13143224

**Published:** 2020-07-20

**Authors:** Beata Kaczmarek

**Affiliations:** Department of Biomaterials and Cosmetics Chemistry, Faculty of Chemistry, Nicolaus Copernicus University in Toruń, Gagarin 7, 87-100 Toruń, Poland; beata.kaczmarek@umk.pl

**Keywords:** phenolic acids, tannic acid, antiviral, antibacterial

## Abstract

As a phenolic acid, tannic acid can be classified into a polyphenolic group. It has been widely studied in the biomedical field of science because it presents unique antiviral as well as antibacterial properties. Tannic acid has been reported to present the activity against Influeneza A virus, Papilloma viruses, noroviruses, Herpes simplex virus type 1 and 2, and human immunodeficiency virus (HIV) as well as activity against both Gram-positive and Gram-negative bacteria as *Staphylococcus aureus*, *Escherichia coli*, *Streptococcus pyogenes*, *Enterococcus faecalis*, *Pseudomonas aeruginosa*, *Yersinia enterocolitica*, *Listeria innocua*. Nowadays, compounds of natural origin constitute fundaments of material science, and the trend is called “from nature to nature”. Although biopolymers have found a broad range of applications in biomedical sciences, they do not present anti-microbial activity, and their physicochemical properties are rather poor. Biopolymers, however, may be modified with organic and inorganic additives which enhance their properties. Tannic acid, like phenolic acid, is classified into a polyphenolic group and can be isolated from natural sources, e.g., a pure compound or a component of a plant extract. Numerous studies have been carried out over the application of tannic acid as an additive to biopolymer materials due to its unique properties. On the one hand, it shows antimicrobial and antiviral activity, while on the other hand, it reveals promising biological properties, i.e., enhances the cell proliferation, tissue regeneration and wound healing processes. Tannic acid is added to different biopolymers, collagen and polysaccharides as chitosan, agarose and starch. Its activity has been proven by the determination of physicochemical properties, as well as the performance of in vitro and in vivo studies. This systematics review is a summary of current studies on tannic acid properties. It presents tannic acid as an excellent natural compound which can be used to eliminate pathogenic factors as well as a revision of current studies on tannic acid composed with biopolymers and active properties of the resulting complexes.

## 1. Introduction

Human organisms are exposed to different external factors which may cause diseases and which pose a threat to people’s lives. According to US National Institute of Health reports, 80% of all microbial infections in the human body are associated with pathogen biofilm formation. There is an increased need to search for effective compounds which would enable protection against pathogens. Since protection against viruses and bacteria is a crucial issue for humans, one of the main healthcare problems is to find effective compounds demonstrating antiviral and antibacterial properties [1,2,3]. Moreover, the evaluation of their effectiveness in real-time use and the examination of their exact application conditions is also essential.

One strategy is related to inhibiting the microbes’ adhesion to surfaces which have been specially modified to repel pathogens [4]. Another strategy is to add antimicrobial compounds as additives during material fabrication or as coatings. Moreover, increasing the material roughness can be effective, as it prevents bacteria attachment to the cell wall. To protect the surface against pathogens, different drugs may be added to the material during fabrication and then released to surroundings. Such a strategy is more effective than traditional drug treatment, i.e., administering pills or injections; however, the problem of bacteria resistance to antibiotics has recently become serious [5,6,7].

There is a growing interest in compounds which may be extracted from natural sources and which have unique antimicrobial properties. Their application has been considered in the form of diet supplements with or as raw materials, i.e., for medical or packaging purposes. Within the last few years, polyphenols have also attracted significant attention and they are being tested for their antiviral and antibacterial properties.

Biomaterials are one of the most rapidly developing fields of science [8,9,10]. There is a growing interest in naturally derived compounds which may be isolated from natural sources and then used as raw compounds for biomaterials preparation. Materials based on both types of natural polymers, proteins, and polysaccharides have been found as biocompatible and non-toxic for the human body [11,12,13]. Thereby, they may be applied as implants, wound dressings, metal coatings, etc. Despite their excellent biocompatibility, the main issue is that the physicochemical properties of biopolymers are rather poor. Moreover, natural polymers may be easily infected by microbes, as they do not possess antimicrobial activity themselves [14]. Therefore, it is necessary to improve biopolymers’ physicochemical properties as well as provide antimicrobial activity by their modification.

Special interest in the biopolymers cross-linkers has been focused on natural compounds. If they contain hydrophilic groups able to form hydrogen bonds, they may be potentially studied as biopolymers cross-linkers. Effective and safe modifiers such as polyphenolic acids have been studied in recent years [15].

Polyphenols constitute a large group of organic compounds, covering a wide range of complex structures. They contain numerous phenolic rings in their structures, with prevalent carboxylic and hydroxyl groups. They may be divided into two groups—phenolic acids and phenolic alcohols. Over seven hundred polyphenolic compounds have already been identified as derivatives from natural sources. They are biosynthesized naturally by plants and marine organisms, from which they are commonly extracted. Polyphenols include flavonoids, phenolic acids, stilbenes, and lignans. As a group, they hold a special position in biological science for their unique biological properties [16].

Phenolic acids support human health protection against chronic degenerative ailments [17]. They are known to present preventive properties against many diseases, e.g., cardiovascular disease, osteoporosis, neurogenerative disease, diabetes mellitus, and even against cancers. Moreover, polyphenols demonstrate active properties against the cells’ metabolic process, as they may block cell propagation and apoptosis [18,19].

The use of natural compounds such as polyphenolic acids is a novel, economical, simple to use, and environmentally friendly approach to health issues. Tannins are a group of polyphenols which commonly occur in nature. They may be easily extracted from plants. The extraction method which is employed in the case of tannins influences the chemical nature of the compounds, their molecular weight, and contamination. Analyzing the isolation conditions, choosing the type of a plant (as well as a plant part i.e., leaves or roots, season), as well as a solvent, deciding on the number of repeating series for final extraction, etc. are also essential. Therefore, it is difficult to find extraction methods which would give the same resulting compounds. It suggests that each extraction method should be followed by the final product characterization [20,21].

## 2. Tannic Acid

Tannic acid (TA; Figure 1) is a natural tannin from the phenolic acid group and consists of a central glucose unit and ten gallic acid molecules attached to it [22]. It may be isolated from both herbaceous and woody types of plants [23]. TA is one of the main examples of tannins which can be efficiently extracted from natural sources with high efficiency, and thus, it attracts much scientific interest. Moreover, it has a higher molecular weight than, for instance, gallic acid. As a result, it has been studied as a biopolymer cross-linker, or an active additive to metals coatings and nanoparticles.

Tannic acid has many unique properties. It has antimutagenic and antitumor properties. Tannic acid shows activity against microorganisms (bacteria and viruses). It acts also as an antioxidant and homeostatic agent. Moreover, tannic acid can neutralize free radicals which cause different diseases’ development such as allergies, diabetes, Parkinson’s, Alzheimer’s, and cardiovascular. Also, it has been proved that tannic acid has anticancer activity. Currently, tannic acid is also being studied as an organic polymer additive, because it reveals bioactive properties and enhances the properties of materials for biomedical applications. Thereby, it is an interesting active compound which may be used as an ingredient in nutritional products, and also various types of consumables [24,25,26,27,28,29,30,31,32,33].

Nowadays, it is especially important to search for natural compounds which are biocompatible and show antiviral and antibacterial activity to protect human organisms against pathogenic factors. Natural compounds may be considered as promising ones to support the fight against many diseases. Tannic acid can be offered as a valuable component of supplements as well as different types of useful materials. In this review, however, I would like to emphasize the antiviral and antibacterial properties of tannic acid, which seem to be of great significance, especially in the time of the COVID-19 pandemic, which has adversely affected human lives, and its consequences show how important it is to carry out studies aimed at health protection.

## 3. Antiviral Activity

Various tannins are often found in plant extracts, and they may differ in their antiviral activity [34]. Such activity has been demonstrated in the case of tannic acid, for which it depends on its molecular weight as well as the extraction method and conditions during the process. There are several studies which prove the antiviral activity of tannic acid (Table 1).

Tannic acid activity against Influenza A virus is 12 times higher than another phenolic acid, gallic acid. The total number of galloyl residues determines its antiviral activity. TA (high molecular weight tannin) activity is related to the inhibition of both the influenza A virus (IAV) receptor binding and neuraminidase activity. Gallic acid, as a low molecular weight tannin, inhibits neuraminidase but not hemagglutination. Thereby, tannic acid exhibits higher activity against IAV [34].

It has been reported that a tannic acid-enriched extract inhibits human Papilloma virus (HPV) type 16 infection. HPV is a non-enveloped type of virus which can cause genital warts or cervical carcinoma. A vaccine against HPV is available; however, it protects only against a minor fraction of over 100 serotypes. High vaccination cost considerably limits the frequency of its use by humans. Thereby, there is a huge demand for cheap and effective active compounds against HPV. TA may bind the HPV host cell receptor and, as a result, inhibit its attachment [34].

Tannic acid is a component of many traditional Chinese medicaments. Its effectiveness has long been known against noroviruses (NoVs). TA inhibits norovirus binding to HBGA receptors. In this study, different forms of hydrolysable tannins were tested. As research shows, TA has the strongest inhibitor which limits the NoVs proteins binding to their HBGA receptors [35].

Herpes simplex virus type 1 (HSV-1) infections are very common, and the virus is an important human pathogen. Medicinal plants have been used for many years for treating human diseases. It was proven that herbal extracts containing tannic acid show activity against HSV-1 in in vitro studies and have low cytotoxicity. Tannic acid inhibits HSV-1 replication, as indicated by the relative absence or reduction of CPE. Most antiviral drugs are toxic, and hence the benefits connected with tannic acid application should be emphasized [36].

Tannic acid was also tested as a silver nanoparticle modifier in effective herpes virus infection treatment [37]. The results confirmed the ability of hydrogels with silver nanoparticles modified by TA to affect viral attachment, impede penetration and cell-to-cell transmission, although profound differences in the activity displayed by the tested preparations toward Herpes simplex virus type 1 (HSV-1) and type 2 (HSV-2) were noted. The effectiveness was also tested in in vivo conditions [37]. Such studies provide pre-clinical assumptions that tannic acid-based materials may be applied against viruses. The antiviral effects of tannic acid–modified silver/copper nanoparticles against HSV-2 by in vitro and in vivo methods were also confirmed, using a murine model of a HSV-2 genital infection [38,39].

Tannins were also studied against some viruses at various stages of infection by determining their influence on the viruses’ replication. TA was studied against human immunodeficiency virus (HIV) as it inhibits HIV replication in H9 lymphocytes [40]. Hydrolysable tannins were studied against HIV by Xu et al. [41] The inhibition effect against human immunodeficiency virus (HIV)-1 protease was confirmed [42]. Uchiumi et al. [43] also studied *Reaumuria hirtella* and *Quercus coccifera* extracts which contain tannic acid.

Tannic acid shows high antiviral effectiveness. It has been studied as a component of extracts isolated from natural sources, but also as a pure compound. Its antiviral activity is based on the virus cell membrane adsorption, which results in inhibiting the virus activity and the ability to attack human cells. However, tannic acid is not commercially registered as a supplement or a drug. It suggests that further studies have to be carried out. In vivo studies on animals showed promising results, however, there is a lack of clinical evidence concerning tannic acid activity. It is difficult to carry out phenolic acid studies because it has a high ability to bind proteins. Thereby, before tannic acid is considered as a potential antiviral compound, its influence on various proteins present in the human body has to be determined.

## 4. Antibacterial Activity

Tannic acid has drawn significant interest, owing to its broad spectrum of chemical and biological properties. The rapid spread of multidrug-resistant bacteria has influenced demand for effective antimicrobial agents which reveal more direct bactericidal mechanisms [44]. Antibiotic resistance is one of the main challenges in antibacterial testing. It leads to higher medical costs, prolonged hospitalization, and increased mortality rate. An expanding list of infections (i.e., pneumonia, tuberculosis, blood poisoning, gonorrhea, and foodborne diseases) shows that antibiotic treatment is becoming more difficult, and sometimes impossible, as antibiotics are becoming less effective. There is a need for new natural antibacterial compounds; also, a better understanding of the mechanism of their actions on bacteria is important.

The antimicrobial activity has been demonstrated for many tannins extracted from plants (Table 2). Tannin-rich plant extracts have shown high antimicrobial effects. Their antibacterial activity depends on conditions such as pH, temperature, type of solvent/matrix, and action time [45,46]. Tannins are multidentate ligands which may bind to proteins, mainly by hydrophobic interactions and hydrogen bonds [34,47]. As a result, the inhibition of bacteria metabolism is achieved.

Dabbaghi et al. [23] have reported the tannic acid activity against *Staphylococcus aureus* and *Escherichia coli* which dependend on the phenolic hydroxyl groups content. Tannic acid was used as a polymer cross-linker, and final hydrogels showed antibacterial activity against both types of bacteria in the case of which the increased activity was related to increasing content of TA in the material.

Anti-infectious properties have also been demonstrated for tannins isolated from green tea extract. They are active against *Streptococcus pyogenes*, which was discussed in the paper by Hull Vance et al. The results obtained during studies indicated that the extract addition inhibited the attachment of the bacteria to the kidney epithelial cells in a dose dependent manner [48].

The antibacterial activity of tannins obtained by extraction from *Anthemis praecox Link* was studied by Belhaoues et al. The tannic acid showed a broad spectrum of activity, especially against *Staphylococcus aureus* and *Enterococcus faecalis*, which suggests that Gram-positive bacteria were most susceptible to tannic acid than Gram-negative ones. Tannic acid functions as an inhibitor of the NorA efflux pump, which is considered as the main mechanism responsible for its antibacterial activity [49].

An antibacterial effect was also detected for *Quercus infectoria* galls extract, which contains tannic acid as the main phenolic compound. A gel containing the extract was prepared by mixing it with cholesterol and soy lecithin. The obtained forms showed antibacterial efficiency against *P. aeruginosa* and *S. aureus*. However, the results of long-term preclinical studies need to be confirmed by further investigation [50].

Tannic acid is a component of *Neolamarckia cadamba* fruits extracts, which demonstrate antibacterial effectiveness against many types bacteria, such as *E. coli*, *P. aeruginosa*, *Y. enterocolitica*, *S. aureus*, *B. cereus*, and *L. innocua*. Tannic acid content was determined as one of the main components in the obtained extracts. The study results suggested that at higher concentrations, the extracts inhibited the sugar and amino acid uptake, which is one of the main mechanisms of bacterial growth inhibition [51].

Tannic acid antibacterial activity has been proven on Gram-positive and Gram-negative bacteria. As Gram-negative bacteria have been a great challenge to modern medicine, the reported TA activity against them has been of most significance. However, there is a lack of preclinical and clinical studies of the effectiveness of tannins against bacteria. Only such research would provide complete data concerning their influence on bacteria cells in the presence of normal human cells. In case the compound showed antibacterial effectiveness, a risk would exist that it would show cytotoxicity to somatic cells. Therefore, it is important to search for compounds presenting antibacterial effectiveness which would not be toxic to human cells. In vivo studies would show a wide range of living cells and tissues responses to tannic acid. Moreover, the examination of bacteria strains which pose the highest risk of infection is important.

## 5. Mechanism of Antimicrobial Activity

Tannic acid activity against viruses is related to the inhibition of receptor binding and the influence on their activity. As it is binds to the cell receptor, it inhibits viruses’ attachment to the different types of surfaces. Moreover, it inhibits the attachment of proteins to the cells which are necessary for the metabolite processes [34,47,51].

Most bacteria can be broadly classified as Gram-positive or Gram-negative. Gram-positive bacteria have cell walls composed of thick layers of peptidoglycan. Gram-positive cells stain purple when subject to a Gram stain procedure. Gram-negative bacteria cell walls have a thin layer of peptidoglycan. Gram-positive bacteria are easier to kill. Gram-negative bacteria are not destroyed by certain detergents which easily kill Gram-positive bacteria [23].

The antibacterial effectiveness of tannins is explained by their ability to pass through the bacterial cell wall up to the internal membrane, interference with the metabolism of the cell, and - as a result—their destruction. In Gram-positive bacteria, the activity of tannins is rapid. However, in Gram-negative bacteria, it is slower as a result of the bilayered membrane presence. Gram-negative bacteria are more harmful and cause certain diseases; so, the examination of this group of bacteria is especially required. Tannic acid has been studied against different types of bacteria so far, both, Gram-positive (mainly *Staphylococcus aureus*) as well as Gram-negative ones (mainly *Escherichia coli*) [23,34,47,48,49].

Tannic acid inhibits the bacteria attachment to the surfaces [48]. A lack of bacteria adhesion to the surface results in bacteria cell death. Moreover, the sugar and amino acid uptake are inhibited by tannic acid what limits the bacteria growth [51]. However, phenolic acid activity against bacteria depends on its concentration, pH, temperature, and type of matrices to which tannic acid was added. Therefore, all types of composites have to be examined in antimicrobial studies [45,46].

## 6. Tannic Acid-Based Biomaterials

Tannic acid has been study as additive to produce biomaterials (Table 3). Rheological measurements of tannic acid-collagen complexes have been reported as hydrogel formation studies [52]. Hydrogels revealed overall pseudoplastic rheological behavior. The formed hydrogels showed viscoelastic behavior that prevails over the viscous contribution, as shown from oscillatory rheometric results. Their high viscosity ensures adhesion to wound and their elasticity prevents against the material damage during application.

Tannic acid-collagen hydrogels studies also presented a significant increase in the antioxidant activity of hydrogels with TA, in comparison to pure collagen. Tannic acid release from such hydrogels was examined. In the case of matrices, the release is faster, which is typical of porous structures. TA released from hydrogels is delivered into the direct spot from the material [53].

Tannic acid-collagen hydrogels were studied with estrogen receptor-positive breast cancer cells, triple-negative breast cancer cells, and normal breast epithelial cells [54]. Recepto-positive breast cancer cells were characterized as more sensitive to TA influence. The fact that released TA induced caspase-mediated apoptosis makes another interesting observation. Tannic acid-collagen complexes have also been successfully studied against A375 melanoma cancer cells [55]. The presence of tannic acid inhibits the melanoma cancer cells growth. Such studies provide a potential for further studying the anticancer properties of tannic acid incorporated into biopolymers.

TA-collagen complexes may also be formed by microwave heating. A thermal analysis was used to determine the tannic acid influence on collagen-based materials. The results showed the higher hydrothermal stability of the cross-linked collagen than that without TA which results from stronger TA to collagen bonding [56].

Tannic acid was studied as a cross-linker for a cell-laden collagen scaffold fabricated via cell-printing. TA addition resulted in both improved scaffold mechanical properties as well as its cellular preosteoblasts activity. Such studies provide evidence that materials based on tannic acid-collagen complexes can be obtained by bio-printing [57].

Tannic acid contains many hydroxyl groups which may cause hydrogen interactions with amine groups of chitosan [58]. It is the basis for the cross-linking process which leads to the improvement of chitosan-based materials properties. TA-chitosan complexes are used to obtain thin solid films. After tannic acid addition, the material’s structure is changed into an anhydrous crystalline conformation when compared to a neat chitosan film. The presence of tannic acid improves the mechanical properties of the films and decreases the degradation rate [59,60]. Moreover, TA added to chitosan improves cell viability, which was determined by carrying out tests by seeding cells on the thin film surface [61].

Tannic acid addition to chitosan results in the decrease of bacteria adhesion, as well as the therapeutic release of phenolic acid into its surrounding at the pH = 7.4 [62]. Thereby, such mixtures may be proposed as coatings, e.g., to cover a metal surface or act as antibacterial protection during implantation.

Tannic acid-chitosan films were tested as drug delivery systems (i.e., doxorubicin hydrochloride) for anticancer treatment [63]. The abundant carboxylate groups in such a mixture increased the loading amount of the drug and decreased its rapid release. The concentration of the released compounds may thereby be modified by the addition of tannic acid, which acts as a cross-linker.

Tannic acid was also tested as a chitosan cross-linker in the hydrogel form [64]. Chitosan molecular weight is an important factor. Hydrogels based on medium molecular weight chitosan showed a reduced degree of swelling when compared to those containing high molecular weight. High molecular weight chitosan has polymeric chains longer than the medium weight one. In such a case, tannic acid has a lower ability to bind functional groups which are more distant in the chain. As a result, the chitosan of medium molecular weight is vulnerable to the cross-linking process. Moreover, the higher tannic acid content in the hydrogel composition results in a higher crosslinking density in the hydrogels and reduces the swelling degree.

Agarose/tannic acid hydrogel scaffolds were fabricated for drug delivery purposes [65]. Tannic acid was studied in release tests, where its concentration was determined in the dependence on the medium pH. The prepared hydrogels showed anti-microbial and anti-inflammatory properties as well as a lack of cytotoxicity. As a result, the proposed hydrogels may be studied by in vivo methods, as they are promising for wound dressing applications.

Agarose-based hydrogels with tannic acid addition show improved mechanical properties than those without TA addition. The wound healing process is stimulated by the tannic acid presence. Moreover, hydrogels are characterized by high biocompatibility [65].

Agarose functionalized by tannic acid was tested as titanium, stainless steel, and silicon coating via direct adsorption [66]. Such coating effectively reduces the adsorption of bovine serum albumin and the adhesion of *Escherichia coli* and 3T3 fibroblasts. It is a promising modification of metal surfaces, aimed at the enhancement of their biological properties.

Starch may interact with phenolic acids by non-covalent bonding formation. A detailed mechanism of interactions has been suggested by Zhu [67]. The complexes were formed by hydrogen interactions between phenolic acid and a polymeric chain. The obtained forms had highly intermolecular, cross-linked, and gel-like network structures; also, the TA content was lower inside and higher outside the material. Such complexes are not homogeneous, which is caused by competing interactions with water molecules. Studies of the pH dependence should be carried out, since it would allow fabricating homogeneous structures for potential biomedical applications [68].

Within the past years, few novel papers of tannic acid-starch complexes have been published. Starch is difficult to modify because of its low solubility in polar solvents. It has significant biological properties; however, more studies of starch-based materials cross-linked by tannic acid should be carried out. Such materials present great potential for biomedical applications, but they also need to be studied in in vitro and in vivo tests.

Hyaluronic acid-based hydrogels undergo rapid degradation processes which limit the range of their applications. Tannic acid has been studied as a hyaluronic acid physical cross-linker in a hydrogel form [69]. It interacts by hydrogen bonding and enhances the material physicochemical properties. The inhibition of degradation by hyaluronidase was noticed. Moreover, for such hydrogels, an increase in cell adhesion to the surface and their proliferation was observed, with no sign of cytotoxicity. The prepared hydrogels possess also antioxidant properties. Significant enhancement of hyaluronic-acid based hydrogels by tannic acid addition may provide great potential for extending the scope of their biomedical applications [70].

Tannic acid was used to form a complex with silk by non-covalent interactions. Gel-like forms based on silk fibroin cross-linked by tannic acid showed improved wet-adhesive properties and stability [71,72]. TA and silk sericin may be conjugated via hydrogen bonding interactions. The mixture may be then deposited on the titanium (Ti) surfaces through surface adhesive trihydroxyphenyl groups in TA [73]. The modified Ti surface showed good protein repellent as well as platelet, bacterial anti-adhesive properties, and low cytotoxicity. Tannic acid-silk hydrogels presented antibacterial efficiency against *S. aureus*, *Candida albicans*, *Cornebacterium,* and *E. coli*. In vivo studies confirmed that applying such hydrogels significantly accelerates the wound healing process.

## 7. Conclusions

Tannic acid is a naturally derived compound which has attracted scientific interest owing to its unique antimicrobial properties. Tannic acid is an interesting compound studied due to its antiviral as well as antibacterial effectiveness. Activity against different viruses, i.e. Influeneza A, Papilloma, noroviruses, Herpes simplex type 1 and 2, and human immunodeficiency virus (HIV) were also reported. Moreover, TA showed activity against Gram-positive and Gram-negative bacteria e.g., *Staphylococcus aureus*, *Escherichia coli*, *Streptococcus pyogenes*, *Enterococcus faecalis*, *Pseudomonas aeruginosa*, *Yersinia enterocolitica, Listeria innocua*.

To sum up, tannic acid is an interesting natural compound with noteworthy antiviral and antibacterial properties confirmed by in vitro methods. However, there is a lack of preclinical and clinical examination results of TA effectiveness in real-time studies. If in vivo studies confirmed its activity, a great opportunity for a large scale industrial use would emerge, mainly for biomaterials fabrication.

So far, tannic acid has been tested in combination with collagen, chitosan, starch, agarose, hyaluronic acid, and silk. In each case, it acts as a cross-linker; however, it is also released from material and may influence the course of medical treatment. Tannic acid has antimicrobial, antioxidant, and anticancer properties. However, its release rate has to be studied to exclude its toxicity, which is correlated with its concentration. Further in vivo studies are required to exhibit multifunctional response on implanted materials with tannic acid, where the local release occurs. Novel studies showed an excellent ability of tannic acid to bind to biopolymers, but what is more important, is that its biological properties have been proven. Recent studies put much attention to its potential application in anticancer treatment; however, such tests should also be performed on cancer tissues. Tannic acid may be called the cross-linker of the future because of its significant novel properties.

## Figures and Tables

**Figure 1 materials-13-03224-f001:**
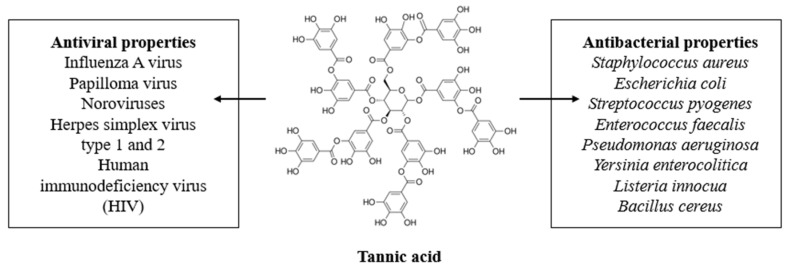
The tannic acid structure and its antimicrobial properties.

**Table 1 materials-13-03224-t001:** Summary of tannic acid antiviral activity studies.

Virus Type	Type of Material/Composition	Reference
Influenza A virus	*Hamamelis virginiana L*. leaf extract	Theisen et al. 2014 [34]
Papilloma virus	*Hamamelis virginiana L*. leaf extract	Theisen et al. 2014 [34]
Noroviruses	Hydrolysable tannins	Zhang et al. 2012 [35]
Herpes simplex virus type 1	*Quercus persica L*. extract	Karimi et al. 2013 [36]
	Green tea extract	Nance et al. 2003 [42]
Herpes simplex virus type 1 and 2	Silver nanoparticles modified by tannic acid in hydrogel form	Szymańska et al. 2018 [37]
Human immunodeficiency virus (HIV)	Tannins testing	Nonaka et al. 1999 [40]
	Hydrolyzable tannins	Xu et al. 2000 [41]
	*Reaumuria hirtella* and *Quercus coccifera* extract	Uchiumi et al. 2003 [43]

**Table 2 materials-13-03224-t002:** Summary of tannic acid antibacterial studies.

Bacteria Type	Type of Material/Composition	Reference
*Staphylococcus aureus*	Tannic acid in polymeric matrix	Dabbaghi et al. 2019 [23]
	*Anthemis praecox Link* extract	Belhaoues et al. 2020 [49]
	*Quercus infectoria* galls extract	Suzilla et al. 2020 [50]
	*Neolamarckia cadamba* fruits extracts	Pandey et al. 2018 [51]
*Escherichia coli*	Tannic acid in polymeric matrix	Dabbaghi et al. 2019 [23]
	*Neolamarckia cadamba* fruits extracts	Pandey et al. 2018 [51]
*Streptococcus pyogenes*	green tea extract	Hull Vance et al. 2011 [48]
*Enterococcus faecalis*	*Anthemis praecox Link* extract	Belhaoues et al. 2020 [49]
*Pseudomonas aeruginosa*	*Quercus infectoria* galls extract	Suzilla et al. 2020 [50]
	*Neolamarckia cadamba* fruits extracts	Pandey et al. 2018 [51]
*Yersinia enterocolitica*	*Neolamarckia cadamba* fruits extracts	Pandey et al. 2018 [51]
*Listeria innocua*	*Neolamarckia cadamba* fruits extracts	Pandey et al. 2018 [51]
*Bacillus cereus*	*Neolamarckia cadamba* fruits extracts	Pandey et al. 2018 [51]

**Table 3 materials-13-03224-t003:** Summary of tannic acid-based biomaterials—their properties and applications.

Polymer Type	Properties	Applications	Reference
collagen	Pseudoplastic rheological behavior, high viscosity	Wound dressings, drug delivery	Brazdaru et al. 2015 [52] Albu et al. 2009 [53]
	Inhibition the melanoma cancer cells growth	Biomaterials with anticancer properties	Ngobili et al. 2015 [54] Bridgeman et al. 2018 [55]
	High hydrothermal stability	Wound dressings	Wu et al. 2018 [56]
	Improved scaffold mechanical properties and cellular preosteoblasts activity	Tissue regeneration	Lee et al. 2018 [57] An et al. 2019 [58]
chitosan	Improved mechanical properties, decreased degradation rate, improved cells viability	Wound dressings	Rubentheren et al. 2015 [59] Kaczmarek et al. 2019 [60] Kaczmarek et al. 2020 [61]
	Decrease of bacteria adhesion	Wound dressings, coatings	Kumorek et al. 2020 [62]
	The medication of released rate by the amount of tannic acid	Drug delivery in anticancer treatment	Sun et al. 2020 [63]
	Reduced swelling degree	Wound dressings	Popa et al. 2018 [64]
agarose	Improved mechanical properties, stimulated wound healing, high biocompatibility	Drug delivery, Wound dressings	Ninan et al. 2016 [65]
	Reduced adsorption of bovine serum albumin and the adhesion of *Escherichia coli*, high biocompatibility	Titanium, stainless steel, and silicon coating	Xu et al. 2017 [66]
Starch	Improved physicochemical properties	Wound dressings	Zhu, 2015 [67] Wei et al. 2019 [68]
Hyaluronic acid	Enhanced physicochemical properties, inhibition of degradation by hyaluronidase, increase in cells adhesion to the surface and their proliferation, antioxidant properties	Wound dressings, tissue regeneration	Lee et al. 2018 [69] Grabska et al. 2019 [70]
Silk	Improved wet-adhesive properties and stability, low cytotoxicity, antibacterial efficiency against *S. aureus*, *Candida albicans*, *Cornebacterium* and *E. coli*	Coatings, tissue regeneration	Gao et al. 2020 [71] Jing et al. 2019 [72] Cheng et al. 2020 [73]

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
