# Peer review of "Tannic Acid with Antiviral and Antibacterial Activity as A Promising Component of Biomaterials—A Minireview"

_materials, 2020, doi:10.3390/ma13143224_

Round 1

Reviewer 1 Report

the author present in this mini review a nice and quite sound overview on the use of tannic acid as antiviral and antibacterial agents. the litterature is quite recent (most of the papers are issued from the last 10 years, with some specific 2020 papers).the review can be published in materials after minor corrections listed below:

1/in the introduction, there are only 6 papers cited, it seems quite low for an introduction, especially considering the large field of research described in the introduction. the latter would deserved to be more illustrated, for instance, page 1 l41-45.

2/the same applied for part 2 (tannic acid) references can be addes at the end of the first paragraph, more specifically in the last sentence. it is valid through the all manuscript.

3/in tables 1 and 2, all biological molecules such as plant species or microorganisms should be written in italic, with the first word bearing a upper case and the second a lower case . this is also valid in the main maniscript, for instance line 170 (Streptococcus ..) or line 174 Anthemis prae...) and in the refernce section also see for instance ref 5, 19, ....

4/in addition on table 1 and 2, the reference should have the number of the ref added, simply the name of the first author and the year is not enough, as the reference section is numbered.

5/line 93, IAV (I guess it's Influenza A virus) should be defined the same way HPV is done line 96

6/line 176, "the tannic acid content was determined at 47.64 µg/mg". it is nice to know, but to what can it be compared? per se and alone, i has no meaning.

Author Response

Ref: materials-850033

Title: Antiviral and antibacterial activity of tannic acid – A review

Journal: Materials

Dear Reviewer, 

I would like to thank for the comments to our manuscript submitted to Materials journal for review process. I would like to thank also the Editor that gave me chance to correct my manuscript. All changes made in my manuscript are written in red. Please find below my answer to the comments.

Reviewer #1:

1/in the introduction, there are only 6 papers cited, it seems quite low for an introduction, especially considering the large field of research described in the introduction. the latter would deserved to be more illustrated, for instance, page 1 l41-45.

Thank you for the suggestion. Additional references is added in the introduction. I also added the addition diagram about tannic acid.

2/the same applied for part 2 (tannic acid) references can be addes at the end of the first paragraph, more specifically in the last sentence. it is valid through the all manuscript.

I appreciate your comment. It is now corrected

3/in tables 1 and 2, all biological molecules such as plant species or microorganisms should be written in italic, with the first word bearing a upper case and the second a lower case . this is also valid in the main maniscript, for instance line 170 (Streptococcus ..) or line 174 Anthemis prae...) and in the refernce section also see for instance ref 5, 19, ....

Thank you very much for the comment. It is now corrected.

4/in addition on table 1 and 2, the reference should have the number of the ref added, simply the name of the first author and the year is not enough, as the reference section is numbered.

I appreciate your comment. It is now corrected.

5/line 93, IAV (I guess it's Influenza A virus) should be defined the same way HPV is done line 96

Thank you for the suggestion. It is now spelled out as: the influenza A virus (IAV)

6/line 176, "the tannic acid content was determined at 47.64 µg/mg". it is nice to know, but to what can it be compared? per se and alone, i has no meaning.

Thank you for your comment. I agree that without comparison it has no meaning. Thereby, I decided to remove this statement as it does not provide important data for this review.

Reviewer 2 Report

This is mini-review focusing on tannic acid (TA) and its antiviral and antibacterial properties. 

Strengths: 

a. reviewed recent articles, most of them were published within the past 10 years. 

b. highlighted antiviral properties of TA, indicated significance in the context of COVID-19 pandemic. Also, highlighted needs for toxicological studies as TA could potentially bind to proteins.

c.  Review on antibacterial properties were appropriate.

Weaknesses:

a. Broad-spectrum use of TA should have been presented in a schematic diagram, including the chemical structure of TA.

b. Mechanism of antimicrobial activity should be included in a separate section. A schematic diagram will be helpful.

c. TA antimicrobial activity seems attractive than other phenolic compounds. It has been mentioned in a few places. However, an explanation needed why TA performs better than other analogues (such as gallic acid).

d. major revision is needed as there are many grammatical and typographical mistakes were identified throughout the manuscript.

- Abstract (line 14): “This systematics review is a summary of current studies on tannic acid properties and presents it as an excellent natural compound which can be used to eliminate pathogenic factors." This sentence needs revision. Split into two sentences and remove “and”.

 - line 78, "to protects" should be "to protect"

- line 93, spell out IAV

- line 193, replace "studied" with "studies"

Author Response

Ref: materials-850033

Title: Antiviral and antibacterial activity of tannic acid – A review

Journal: Materials

Dear Reviewer, 

I would like to thank for the comments to our manuscript submitted to Materials journal for review process. I would like to thank also the Editor that gave me chance to correct my manuscript. All changes made in my manuscript are written in red. Please find below my answer to the comments.

Reviewer #2:

  1. Broad-spectrum use of TA should have been presented in a schematic diagram, including the chemical structure of TA.

Thank you for the suggestion. I added the schematic diagram about tannic acid (Figure 1).

  1. Mechanism of antimicrobial activity should be included in a separate section. A schematic diagram will be helpful.

Thank you for the suggestion. It is now pointed in separate section (point 5).

  1. TA antimicrobial activity seems attractive than other phenolic compounds. It has been mentioned in a few places. However, an explanation needed why TA performs better than other analogues (such as gallic acid).

I appreciate your suggestion. The main advantage of tannic acid compared to gallic acid is its higher molecular weight. As a result, tannic acid is more effective additive than gallic acid in materials science. Also, tannic acid  may be extracted from natural sources with high efficiency. It is now mentioned in the paper:

“TA is one of the main examples of tannins which can be efficiently extracted from natural sources with high efficiency, and thus, it attracts much scientific  interest. Moreover it has higher molecular weight than for instance gallic acid. As a result, it has been studied as a biopolymer cross-linker, or an active additive to metals coatings and nanoparticles”.

Moreover, it tannic acid has higher antiviral activity:

“Tannic acid activity against Influenza A virus is 12 times higher than another phenolic acid, gallic acid.”

  1. major revision is needed as there are many grammatical and typographical mistakes were identified throughout the manuscript.

I appreciate your comment. Manuscript was read again and grammatic mistakes were corrected.

- Abstract (line 14): “This systematics review is a summary of current studies on tannic acid properties and presents it as an excellent natural compound which can be used to eliminate pathogenic factors." This sentence needs revision. Split into two sentences and remove “and”.

Thank you for the suggestion. It is now corrected:

“This systematics review is a summary of current studies on tannic acid properties. It presents tannic acid as excellent natural compound which can be used to eliminate pathogenic factors.”

 - line 78, "to protects" should be "to protect"; - line 193, replace "studied" with "studies"

I appreciate you comment. It is now corrected.

- line 93, spell out IAV

It is now spelled out as: the influenza A virus (IAV)

Round 2

Reviewer 2 Report

The quality of the revised manuscript has been partly improved. 

Mechanism of antimicrobial activity (Section 5; lines 268-325) has been added. Unfortunately, no references have been cited. This section must include appropriate citations. See below. 

Lines 268 - 308: citation needed.

Lines 309 - 313: citation needed.

Lines 314 - 320: citation needed.

Lines 321 - 325: citation needed.

Careful review of the manuscript draft is needed. For example..

Line 321: "Lack bacteria adhesion to the surface". "Lack bacteria...." should be replaced with "Lack of bacteria...". 

Section 6 (Tannic acid-based biomaterials): Addition of this section is aligned with author's background expertise. However, the minireview focus appears to be partly compromised. Is this reviewing author's earlier work on biomaterials or the antimicrobial aspect of tannic acid? The title of this manuscript is now confusing. I suggest that this section can be summarized into one paragraph, keeping the focus on antiviral and antibacterial properties of tannic acid. 

Author Response

Ref: materials-850033

Title: Antiviral and antibacterial activity of tannic acid – A review

Journal: Materials

Dear Reviewer, 

I would like to thank for the comments to our manuscript submitted to Materials journal for review process. I would like to thank also the Editor that gave me chance to correct my manuscript. All changes made in my manuscript are written in yellow. Please find below my answer to the comments.

Reviewer #2: (round 2)

Mechanism of antimicrobial activity (Section 5; lines 268-325) has been added. Unfortunately, no references have been cited. This section must include appropriate citations. See below.

Lines 268 - 308: citation needed.

Lines 309 - 313: citation needed.

Lines 314 - 320: citation needed.

Lines 321 - 325: citation needed.

Thank you very much for the suggestion. Citations are now placed in the manuscript.

Careful review of the manuscript draft is needed. For example..

Line 321: "Lack bacteria adhesion to the surface". "Lack bacteria...." should be replaced with "Lack of bacteria...".

Manuscript was carefully checked. All corrections are now in yellow. I hope that the present form is now acceptable.

Section 6 (Tannic acid-based biomaterials): Addition of this section is aligned with author's background expertise. However, the minireview focus appears to be partly compromised. Is this reviewing author's earlier work on biomaterials or the antimicrobial aspect of tannic acid? The title of this manuscript is now confusing. I suggest that this section can be summarized into one paragraph, keeping the focus on antiviral and antibacterial properties of tannic acid.

Thank you very much for the suggestion. This part of manuscript was added as the academic editor suggested. I summarized now this part into one paragraph.

Round 3

Reviewer 2 Report

Manuscript quality has been improved. I recommend acceptance of the manuscript in its current form.